# INFORMATION-THEORETIC WORLD MODEL LEARNING FOR DENOISED PREDICTIONS

## ABSTRACT

Humans excel at isolating relevant information from noisy data to predict the behavior of dynamic systems, effectively disregarding non-informative, temporally-correlated noise. In contrast, existing reinforcement learning algorithms face challenges in generating noise-free predictions within high-dimensional, noise-saturated environments, especially when trained on world models featuring realistic background noise extracted from natural video streams. We propose a novel information-theoretic approach that learn world models based on minimising the past information and retaining maximal information about the future, aiming at simultaneously learning control policies and at producing denoised predictions. Utilizing Soft Actor-Critic agents augmented with an information-theoretic auxiliary loss, we validate our method's effectiveness on complex variants of the standard DeepMind Control Suite tasks, where natural videos filled with intricate and task-irrelevant information serve as a background. Experimental results demonstrate that our model outperforms eight state-of-the-art approaches in various settings where natural videos serve as dynamic background noise. Our analysis also reveals that all these methods encounter challenges in more complex environments.

## 1 INTRODUCTION

A major open problem in Reinforcement learning (RL) is to learn the dynamics and control policies from the high-dimensional observations such as images (Ha & Schmidhuber, 2018; Lillicrap et al., 2016; Hafner et al., 2020a; 2021a; Hansen et al., 2022). Conventionally, it is assumed that the observations in the environment, often derived through hand-engineered features, consist exclusively of task-relevant information. This allows RL algorithms to operate in a controlled setting with optimal efficiency, primarily due to the absence of exogenous noise (unrelated or uncontrollable external variables such as weather variations or random background movements), that could potentially hinder the learning process.

In the real world, the landscape is vastly different, brimming with a plethora of information, much of which is irrelevant to a specific task. The challenge lies in accurately identifying task-relevant information and avoid the modeling of temporally correlated dynamics of the background noise. Prior RL methodologies (Yarats et al., 2021; Hafner et al., 2020a; Ha & Schmidhuber, 2018) that derive representations directly from observations, often integrate task-irrelevant information into their representations. They struggle to disentangle the noise from relevant information, unnecessarily modeling noise dynamics, leading to sub-optimal performance under noise (see Figure 4).

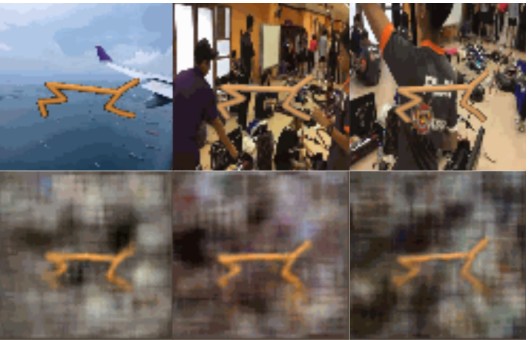

Figure 1: **Top Row:** Ground truth data from a random sequence. **Bottom Row:** Reconstruction from DPI.

The process of computing representations relies on the past inputs, while the imagination and exploration are directed towards future (Hafner et al., 2020b). Our objective is to develop a cohesive

perspective on how an agent formulates its current representation after observing past input and before observing future. Could it be feasible to model this process as an information flow, transitioning from past to future, mediated by the current state?

We introduce Denoised Predictive Imagination (DPI), a model-based reinforcement learning approach that leverages information theory to learn robust and meaningful representations. DPI models Predictive Information (Bialek & Tishby, 1999), the mutual information between the past and the future, and employs the Information Bottleneck principle (Tishby et al., 2000) to derive a compact representation of the current state from historical observations, while preserving maximal predictive information about future outcomes. Essentially, DPI focuses on learning a concise abstraction of the system dynamics and leverages it to learn control policies and generate noise-free future predictions. This is achieved through deriving an objective integrating two central ideas: minimization of mutual information about past and the maximization of predictive ability for future. This dual objective consists of two contrastive losses and is formulated as a Lagrangian optimization problem. DPI outperforms eight existing state-of-the-art models on six modified DeepMind control (DMC) tasks. While in this paper we focus on the algorithmic derivation and the performance of DPI, the information theoretic nature of it enables future investigations of generalization, stability and robustness aspects.

## 2 RELATED WORK

In this section, we delve into related work on reinforcement learning from visual input, focusing specifically on model-based approaches and representation learning concepts. For a more comprehensive discussion, refer to the Supplementary Material.

**Model-based Reinforcement Learning.** These models simultaneously learn policy and transition dynamics, which can be used for planning, and are often sample efficient due to their ability to handle rich observations (Kaiser et al., 2020; Chua et al., 2018; Hafner et al., 2019; Ebert et al., 2018; Lowrey et al., 2019; Gelada et al., 2019; Lee et al., 2020a). World Models Ha & Schmidhuber (2018) uses recurrent latent model to imagine future frames. Stochastic Optimal control with Latent Representations (SOLAR, Zhang et al. (2019)) model dynamics with linear-quadratic regulator. In particular, Dreamer (Hafner et al., 2020a) optimises policies via backpropagating through latent dynamics and uses recurrent state-space model for planning in latent space. These reconstruction-based methods perform effectively in standard environments. However, when exposed to environments with noise distractors, they struggle to bifurcate between information they should reconstruct and what they should disregard.

**Learning Representations and RL.** Recent works (Chen et al., 2020; Henaff, 2020; Tian et al., 2020) have demonstrated progress in learning representations from unlabeled data. These concepts have been integrated into reinforcement learning by works like (Laskin et al., 2020; Oord et al., 2018; Shu et al., 2020; Ma et al., 2021; Oord et al., 2018; Ma et al., 2021; Hjelm et al., 2019). However, the identification and utilization of task-relevant information was not addressed. In contrast, our approach, similar to PI-SAC (Lee et al., 2020b), quantifies and compresses predictive data, excluding irrelevant past information. Yet, unlike strategies such as Dynamic Bottleneck (DB, Bai et al. (2021)) and Sequential Information Bottleneck (SIBE, You et al. (2022)), our approach not only seeks compact representations under noisy conditions, but also emphasizes on achieving noiseless future predictions and treating temporal noise along representations.

**Learning Control from pixels with distractors.** Recent developments in model-based RL (Zhang et al., 2021; Ma et al., 2021; Nguyen et al., 2021; Fu et al., 2021; Bai et al., 2021; You et al., 2022; Efroni et al., 2022; Wang et al., 2022; Islam et al., 2022; Tomar et al., 2023) have put forward a variety of innovative ideas aimed at extracting relevant information from observations. Contrastive Variational Reinforcement Learning (CVRL, Ma et al. (2021)) leverages a contrastive loss to learn representations and dynamics, thereby planning in the latent space by maximizing the MI between observations and latent representations. Deep Bisimulation for Control (DBC, Zhang et al. (2021)) learns control policies by learning representations of the states that preserve the bisimulation metric. Temporal Predictive Coding (TPC, Nguyen et al. (2021)) shares conceptual similarities with our approach, striving to eliminate temporal noise while focusing only on the relevant aspects. More recent methods such as Task Informed Abstractions (TIA, Fu et al. (2021)) maintain two separate latent models, one for tasks and another for distractors, bifurcating noise and signal. TIA falters in

achieving better rewards when the grayscale background is replaced with RGB (see the experimental section). Our work bypasses the need for explicitly defining these types of model rules and instead builds on a general information-theoretic model wherein these types of features implicitly emerge.

## 3 NOTATION AND PRELIMINARIES

**Reinforcement Learning.** An agent operates in a Markov Decision Process (MDP), which is characterised by a tuple $\mathcal{M} = (\mathcal{O}, \mathcal{A}, \mathcal{P}, \mathcal{R}, \gamma)$, consisting of the observation space $\mathcal{O}$ with observations $o$ (we interchangeably use "states" and "observations"), action space $\mathcal{A}$ with actions $a$, transition dynamics $\mathcal{P}$, Reward space $\mathcal{R}$ and discount factor $\gamma \in [0, 1]$. The encoder $\phi(z|o)$ produces a latent representation $z$ from observations, and then the policy $\pi(a|z)$ decodes this latent representation into actions. The goal of RL is to learn a policy $\pi^*(a|z)$ that maximizes the expected cumulative discounted rewards $\mathcal{J}_\pi = argmax_\pi \mathbb{E}_p \left[ \sum_t \gamma^{t-1} r_t \right]$.

**Predictive Information.** Predictive Information (PI) is a quantity that measures how much our observations from the past can inform us about the future Bialek & Tishby (1999) . Mathematically, it can be defined as the mutual information (MI) between the past ($x_{past}$) and the future ($x_{future}$), denoted as $I(x_{past}; x_{future})$. Assuming temporal invariance (any fixed time length is expected to have the same entropy), PI becomes a subextensive quantity, as expressed by $\lim_{T \to \infty} I(T)/T = 0$, where $I(T)$ is the predictive information over a time window of length 2T (with T steps of the past predicting T steps into the future), see Equation 3.1 in Bialek et al. (2001). As the time frame increases, the past contains a diminishing predictive value for the future. In order to capture only the necessary information from $x_{past}$ for predicting $x_{future}$, a compressed representation of $x_{past}$ is required.

**Information Bottleneck.** For learning this compressed representation, we utilize the Information Bottleneck (IB) principle Tishby et al. (2000). IB aims at learning a representation $z$ that aims to optimally compress the information provided by the input $x \in X$, i.e. minimize $I(x; z)$, while still maintaining enough knowledge to predict the outcome $y \in Y$, i.e. maximize $I(z; y)$. This objective is unified with the inclusion of a Lagrangian multiplier and formalized as $max\ I(z; y) - \beta I(x; z)$. The parameter $\beta$ controls the information flow from the input $x$ to the latent representation, balancing the trade-off between information preservation and compression.

## 4 DENOISED PREDICTIVE IMAGINATION

Denoised Predictive Imagination (DPI) is an information theory-based approach, that encapsulates the notions of predictive information and the information bottleneck. This core idea enables the learning of a compressed representation from high-dimensional observations, distilling task-relevant details from past observations, and leveraging this refined knowledge for future predictions while effectively filtering out noise. We hypothesise that the current state should encapsulate the requisite and meaningful information essential to perform the task. If the information is insufficient, the latent representations may fail to capture all the task-relevant information, leading to sub-optimal learning outcomes. On the other hand, if we incorporate an overabundance of information, our representations could become encumbered with noise-related artifacts that results in a dilution of task-relevant data and in a performance decrease.

We denote the latent representations for the past observations by $o_{t-}$, current observation by $o_t$, and the future observations by $o_{t+}$. We use $z_{t-}, z_t$ and $z_{t+}$ respectively for the latent space. For consistency and clarity, we establish that the episode initiates at time $t = 1$ and terminates at the horizon $t = T$. The objective is to encode observations $(o_{t-}, o_t)$ into latent representations $(z_{t-}, z_t)$, transform them to next state representations $z_{t+}$, and decode into future observations $o_{t+}$ (Figure 2). Consequently, this process creates a two-fold bottleneck: one while transforming observations into latent representations and vice-versa ($o_t \leftrightarrow z_t$), and another when acquiring the latent representation itself from other latent representations ($z_{t-1} \to z_t \to z_{t+1}$). In this context, our transition function can be conceptualized as a model operating simultaneously as an encoder and a decoder, encoding $z_t$ from $z_{t-}$ and decoding $z_t$ to yield $z_{t+}$, with bottleneck being $z_t$.

Intuitively, we obtain task-relevant information from raw observations into our latent representations by minimising mutual information $I(o_{t-}, o_t; z_{t-}, z_t)$ while maximising the mutual informa-

tion $I(o_t, o_{t+}; z_t, z_{t+})$, which preserves the predictive information for the reverse scenario. When expressed in Lagrangian formulation, we obtain,

$$\min \ I(o_{t-,t}; z_{t-,t}) - \beta_1 I(o_{t,t+}; z_{t,t+}). \tag{1}$$

In order to learn temporal abstractions and compressed representations from a sequence of past states and acquire relevant predictions, we employ the principle of Information Bottleneck. We apply a Lagrangian on the latent space with the aim of minimising $I(z_{t-}; z_t)$ and maximising $I(z_t; z_{t+})$,

$$\min \ I(z_{t-}; z_t) - \beta_2 I(z_t; z_{t+}). \tag{2}$$

Merging objectives from equation (1) and (2), we obtain a unified Lagrangian optimizing problem,

$$\min \ \Big[ \underbrace{I(o_{t-,t}; z_{t-,t})}_{\substack{\text{Historical} \\ \text{observation model}}} + \underbrace{I(z_{t-}; z_t)}_{\substack{\text{Historical latent} \\ \text{space dynamics}}} \Big] - \Big[ \underbrace{\beta_1 I(o_{t,t+}; z_{t,t+})}_{\substack{\text{Predictive} \\ \text{observation model}}} + \underbrace{\beta_2 I(z_t; z_{t+})}_{\substack{\text{Predictive latent} \\ \text{space dynamics}}} \Big], \tag{3}$$

where $\beta_1$ and $\beta_2$ are the Lagrangian multipliers. This implies that the problem can be optimised by minimizing the upper bound associated with the past, as represented by the first two terms, and simultaneously maximizing the lower bound related to the future, embodied in the final two terms. The objective of our DPI considers action dependencies implicitly through the latent space representations, $p(z_t | z_{t-}, a_{t-})$, thereby reflecting the innate characteristics of system transitions. This compatibility with RL principles facilitates a seamless integration of our approach into existing RL algorithms, where DPI can serve as an auxiliary function, significantly enhancing the learning of representations. Due to space limitations, all subsequent derivations and details are in the Supplementary Material (Section 1).

## 4.1 STATE SPACE MODEL

We use the state-space model described in Figure 2 with,

Encoder Representation: $z_t \sim p_\varphi(z_t \mid o_t)$

Transition dynamics: $z_{t+1} \sim q_\theta(z_{t+1} \mid z_t, a_t, h_t)$

Observation model: $o_t \sim r_\psi(o_t \mid z_t)$

History model: $h_t \sim p(h_t \mid h_{t-1}, a_{t-1}).$ (4)

The conditional $p(h_t \mid h_{t-1}, a_{t-1})$ denotes the history model, that encapsulates the past variables into a single history variable i.e.,

$$h_t = \{z_{t-1}, a_{t-1}, ..., z_1, a_1\},$$
$$= \{z_{t-1}, a_{t-1}, h_{t-1}\}. \tag{5}$$

This is a crucial modelling component that is discussed and used in the next subsections.

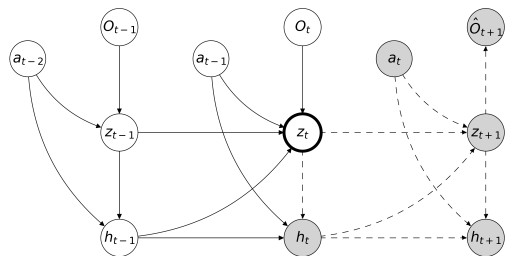

Figure 2: **State-space model.** The variable $z_t$ acts as a bottleneck for the model, serving as a critical link between the historical (white circles) and predictive elements (grey circles). Solid edges designate the inputs required for inference, while the dotted edges represent the generative components.

## 4.2 MINIMISING THE UPPER BOUND OF THE PAST MUTUAL INFORMATION

This subsection discusses the minimization of the first two terms in the Lagrangian of DPI in Equation 3.

**Upper bound of historical latent space dynamics.** We aim at minimising the tractable upper bound on the mutual information $I(z_{t-}; z_t)$. The mutual information can be represented as,

$$I(z_1; ...; z_t) = \mathbb{E}_{p(z_1, ..., z_t)} \left[ \log \frac{p(z_1, ..., z_t)}{\prod_{k=1}^{t} p(z_k)} \right],$$

We incorporate actions into the model by introducing a conditional probability distribution $p(z_{t-}, z_t | a_{t-})$,

$$I(z_{1:t}) = \mathbb{E}_{p(z_{1:t}, a_{1:t-1})} \left[ \log \frac{p(z_{1:t}) p(z_{1:t}|a_{1:t-1})}{p(z_{1:t}|a_{1:t-1}) \prod_{k=1}^{t} p(z_k)} \right] \leq \mathbb{E}_{p(z_{1:t}, a_{1:t-1})} \left[ \log \frac{p(z_{1:t}|a_{1:t-1})}{\prod_{k=1}^{t} p(z_k)} \right] \tag{6}$$

Utilising the chain rule in conditional probability and for every $t$, substituting $\{z_{t-1}, a_{t-1}, h_{t-1}\}$ as $h_t$ like Equation (5), we can write Equation (6) as

$$I(z_{1:t}) \leq \sum_{k=1}^{t-1} \mathbb{E}_{p(z_k, a_k)} \left[ \log \frac{p(z_{k+1}|z_k, a_k, h_k)}{p(z_{k+1})} \right] = \sum_{k=1}^{t-1} I\big(z_{k+1}; z_k, a_k, h_k\big). \tag{7}$$

In essence, this implies that we can optimize the mutual information between the past latent representations and the present state's representation by minimising the upper bound of the MI for each individual, independent transition in a Markovian manner.

For the purpose of minimizing this upper bound, we employ Contrastive Log-ratio Upper Bound of Mutual Information (CLUB, Cheng et al. (2020)), where the core idea is to estimate the MI through the difference of conditional probabilities for positive and negative sample pairs. Since the conditional distribution $p(z_{k+1}|z_k, a_k, h_k)$ is intractable, the upper bound of $I(z_{k+1}; z_k, a_k, h_k)$ cannot be directly minimized. As a consequence, we introduce a variational distribution $q(z_{k+1}|z_k, a_k, h_k)$, serving essentially as the transition function of the model, parameterised by $\theta$, to approximate the upper bound of mutual information,

$$I(z_{k+1}|z_k, a_k, h_k) = \frac{1}{N}\sum_{i=1}^{N} \left[ \log \hat{q}_\theta - \frac{1}{N}\sum_{j=1}^{N} \log \hat{q}_\theta \right] = I_{\text{CLUB}} , \tag{8}$$

where $\hat{q}$ denotes $q_\theta(z_{k+1}^i|z_k^i, a_k^i, h_k^i)$, i.e. the $i$-th sample at $k$-th timestep. We obtain the negative sample pair $(z_{k+1}', (z_k, a_k, h_k))$ via random shuffling.

**Upper bound of the historical observation model.** As in the previous section, it can be shown that an upper bound for $I(o_{1:t}, z_{1:t})$ can be derived by introducing the conditional distribution $p(z_{t-}, z_t | a_{t-})$,

$$I(o_{1:t}; z_{1:t}) \leq \mathbb{E}_{p(z_{1:t}, o_{1:t})} \left[ \log \frac{p(z_{1:t}|o_{1:t})}{p(z_{1:t}|a_{1:t-1})} \right].$$

Taking this further, we employ the same tractable variational distribution drawn from our transition function,

$$I(o_{1:t}; z_{1:t}) \leq \sum_{k=1}^{t-1} \mathbb{E}_{p(z_k, o_k, a_k)} \left[ \log \frac{p(z_{k+1}|o_{k+1})}{q_\theta(z_{k+1}|z_k, a_k, h_k)} \right] = I_{\text{LTC}} . \tag{9}$$

This term is an upper-bound for $I(o_{1:t}, z_{1:t})$, quantifying the ratio between the latent representation derived from the encoder and the transitioning state obtained from a past representation when a specific action is applied. Intuitively, this constrains the latent dynamical model (transition function) to diverge minimally from the latent representations obtained from the observation encoder. Hence, we refer to this term as the Latent Consistency Loss $\mathcal{L}_{\text{LTC}}$.

### 4.3 Maximising the Lower Bound of the Predictive Mutual Information

This subsection discusses the maximization of the last two terms in the Lagrangian of DPI in Equation 3.

**Lower bound of the predictive latent space dynamics.** In order to obtain the lower bound on this MI term, we factorise the transition model by applying the chain rule,

$$
\begin{aligned}
I(z_{t:T}) &= \mathbb{E}_{p(z_{t:T})} \left[ \log \frac{p(z_{t:T})}{\prod_{k=t}^{T} p(z_k)} \right] = \mathbb{E}_{p(z_{t:T})} \left[ \log \prod_{k=t}^{T-1} \frac{p(z_k|z_{k+1:T})}{p(z_k)} \right], \\
&\geq \sum_{k=t}^{T-1} \mathbb{E}_{p(z_k, a_k)} \left[ \log \frac{p(z_k|z_{k+1}, a_k)}{p(z_k)} \right], \\
&= \sum_{k=t}^{T-1} I\big(z_{k+1}, a_k; z_k\big).
\end{aligned}
\tag{10}
$$

The mutual information objective $I(z_{k+1}, a_k; z_k)$ can be decomposed using the chain rule for mutual information, yielding $I(z_k; z_{k+1}) + I(z_k; a_k|z_{k+1})$. The first component, solely depends on state-transitions. It is closely related to the predictive coding objective (Oord et al., 2018; Anand et al., 2019). Omitting actions could impair the model's capability to determine the optimal actions (Rakelly et al., 2021). The second term can be represented in terms of conditional entropy as $H(a_k|z_k) - H(a_k|z_k, z_{k+1})$. The term $H(a_k|z_k, z_{k+1})$ effectively characterizes the entropy of the inverse dynamics, conceptually aligns closely with an extensive spectrum of prior studies that have focused on exploration and unsupervised learning of representations (Zhang et al., 2018; Pathak et al., 2017; Chandak et al., 2019; Bharadhwaj et al., 2022). From an intuitive perspective, inverse models operate as an agreement mechanism between the actual and the ground truth action representations. This mechanism enables the representation to capture only those aspects of the state that are essential for predicting the action, thereby discarding potentially irrelevant information. The MI term in Equation 10 can be viewed as a combined objective that optimises state transitions with the regularization of action representations.

For optimising this lower bound, we will utilise contrastive learning (Oord et al., 2018), which yields a variational lower bound of the mutual information in Equation 10. Strategies employed by He et al. (2020); Laskin et al. (2020) relies on data augmentation to generate positive and negative samples. Contrary to them, we take inspiration from Bai et al. (2021) that incorporate policy transitions to obtain these samples. Positive samples are directly acquired by sampling transitions $(z_t, a_t, z_{t+1})$, while the construction of negative samples involves randomly sampling $z_t^*$ and concatenating it with $(a_t, z_{t+1})$. As a result, we produce samples $(z_t^*, a_t, z_{t+1})$ that deviate from the transition dynamics. Thus we obtain MI objective as,

$$I(z_{k+1}, a_k; z_k) \geq \mathbb{E}_{p,N} \left[ \log \frac{e^{\sigma(z_k, a_k, z_{k+1})}}{\sum_{z_k^* \in N^- \cup z_k} e^{\sigma(z_k^*, a_k, z_{k+1})}} \right] \triangleq I_{\text{NCE}} , \qquad (11)$$

where $N$ is the set of negative samples and $\sigma$ is the score function. Score functions quantifies the similarity between paired examples, providing high score to the positive examples and low score to the negative examples. We opt for bilinear products as our score function (Oord et al., 2018; Laskin et al., 2020; Henaff, 2020), defined as $c(a_t, z_{t+1})^T \mathcal{W} z_t$, where $c(\cdot, \cdot)$ is the concatenation function parameterised by a neural network and $\mathcal{W}$ is the learnable parameter of the score function. The concatenation network combines the action and subsequent latent representation into a single vector, as shown in Figure 3b.

**Lower bound of the predictive observation model.** Directly maximizing $I(z_{t,t^+}; o_{t,t^+})$ is infeasible due to its marginal's intractability. Similar to Alemi et al. (2017), we propose to optimise a lower bound on our MI,

$$I(z_{t:T}; o_{t:T}) = \mathbb{E}_{p(z_{t:T}, o_{t:T})} \left[ \log \frac{p(o_{t:T}|z_{t:T})}{p(o_{t:T})} \right] = \mathbb{E}_{p(z_{t:T}, o_{t:T})} \left[ \log \prod_{k=t}^{T} \frac{p(o_k|z_k)}{p(o_k)} \right],$$

$$\geq \sum_{k=t}^{T} \mathbb{E}_{p(z_k, o_k)} \left[ \log \frac{r_\psi(o_k|z_k)}{p(o_k)} \right],$$

where where $p(o_k|z_k)$ is an intractable conditional distribution and $r_\psi(o_k|z_k)$ is a tractable variational decoder, represented by a neural network with parameters $\psi$. We rule out the entropy term as it is independent of our optimization procedure,

$$I(z_{t:T}; o_{t:T}) = \sum_{k=t}^{T} \mathbb{E}_{p(z_k, o_k)} \left[ \log r_\psi(o_k|z_k) \right] = I_{\text{Rec}} . \qquad (12)$$

$I_{\text{Rec}}$ can be interpreted as the log-likelihood of the observations given the state encodings.

## 4.4 COMBINED OBJECTIVE

Our optimization strategy can be unified into a single objective function as,

$$\min_{\theta, \psi, \phi, \mathcal{W}} \mathcal{L}_{DPI} = [\alpha_1 I_{LTC} + \alpha_2 I_{CLUB}] - [\beta_1 I_{Rec} + \beta_2 I_{NCE}]. \qquad (13)$$

The two losses, $I_{LTC}$ and $I_{Rec}$, are responsible for the representations from the encoder and decoder respectively, while the other two terms, $I_{CLUB}$ and $I_{NCE}$, formulated as a contrastive loss, control the representations of the transition functions. They are jointly optimized.

## 4.5 PRACTICAL IMPLEMENTATION WITH SOFT-ACTOR CRITIC

We jointly train DPI with SAC, an off-policy model-free reinforcement learning method, by incorporating Equation (13) as an auxiliary objective while training the algorithm (Supplementary Material Section 3.1). The transition model, accounting for latent dynamics, is designed to capture the inherent stochasticity of the transitions. It is parameterised with a neural network that returns a Gaussian distribution defined by its mean and variance. The Observation model implemented as a Deconvolutional Neural Network. The History model is implemented as a Gated Recurrent Unit (GRU, Cho et al. (2014)). We utilize a stochastic encoder to obtain representations from the images (Eysenbach et al., 2021; Theis & Agustsson, 2021), parameterised by $\varphi$. For encoding subsequent observations, we leverage an exponential moving average of the online network parameters, denoted as $\varphi_m$ (He et al., 2020). We utilise the same principle for latent targets Hansen et al. (2022) for transition function, as it should ensure more stable learning process, accommodating any potential fluctuations in the learning (Figure 3a). The complete algorithm with SAC is described in the Supplementary material.

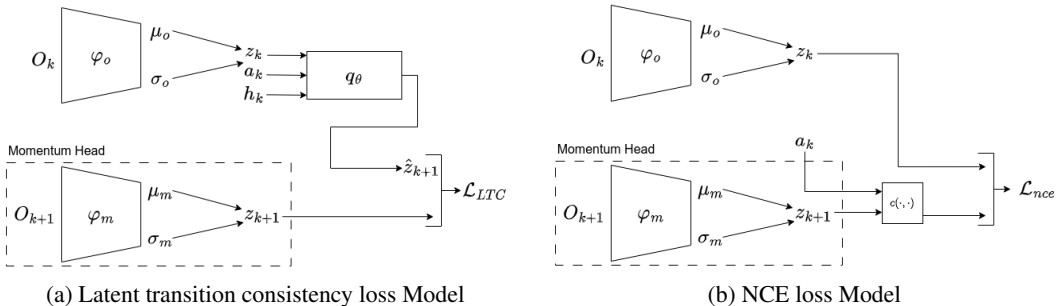

(a) Latent transition consistency loss Model  (b) NCE loss Model

Figure 3: Representation of models used for calculating auxiliary losses (a) LTC loss $\mathcal{L}_{LTC}$ and NCE loss ($\mathcal{L}_{NCE}$). Encoder and target encoder parameters are defined as $\varphi_0$ and $\varphi_m$ respectively. a) Once the current representation is obtained, it is passed through the transition function $q_\theta$ to obtain the next latent representation, from which the $\mathcal{L}_{LTC}$ is finally calculated (Algorithm 2 in Supplementary material). b) Next latent representation and current action is passed via concatenation function $c$ to obtain unified representation, then compared with current state representation via contrastive learning.

## 5 EXPERIMENTS

In this section, we conduct a thorough empirical assessment of the proposed DPI method on the DeepMind control suite (DMC, Tassa et al. (2018)) in various settings and compare it with existing state-of-the-art approaches. We evaluate three distinct types of environments: (i) Standard environment with a static background, (ii) Natural environment with video-based, real-world backgrounds, and (iii) Random environment with varying backgrounds in each frame. To underline the significance of each element in the model, we conclude this section with an ablation study.

### 5.1 ENVIRONMENT SETTINGS

For all three environments, we conducted experiments on six DMC tasks: Cheetah Run, Walker Walk, Cartpole Swingup, Reacher Easy, Pendulum Swingup and Cup Catch. These robot control tasks pose different challenges, such as sparse rewards, contacts and complex dynamics. For the standard settings, no perterbutations are applied to the observations. The observations are RGB images of the size $84 \times 84 \times 3$. By incorporating the ground plane, a substantial portion of the background image is obscured, thereby simplifying the task at hand. Thus, the ground plane is eliminated to maximize the utilization of the background image. These natural videos are incorporated from Kinetics 400 dataset Kay et al. (2017) at random. We used videos from random categories compared to the simplified challenge in DBC Zhang et al. (2021) who only considered the driving category. Contrary to the predominant use of grayscale images in benchmarking, we employing RGB videos in the background. We independently sampled 100 videos separately for training and

testing. More information about the background noise is provided in the Supplementary Material (Section 5.1).

Figure 4: **Natural Background Setting**. Test performance of our method (DPI) and eight baselines on six robot control tasks, with added videos as background noise. Shown is the mean of three runs where shaded areas denote 95% confidence intervals.

## 5.2 BASELINES AND IMPLEMENTATION DETAILS

In this evaluation, we compare our approach to a selection of eight most-closely related approaches i.e. Dreamer (Hafner et al., 2020a), Dreamer-V2 (Hafner et al., 2021b), Task-informed Abstractions (TIA, Fu et al. (2021)), Denoised MDPs (Wang et al., 2022), Deep Bisimulation for Control (DBC, Zhang et al. (2021)), Self-Predicting Representations (SPR, Schwarzer et al. (2021)), Variational Sparse Gating (VSG, Jain et al. (2022)) and Temporal Predictive Coding (TPC, Nguyen et al. (2021)). These selected methods are distinguished by their superior performance and accompanied by publicly accessible source code. The task return is examined every 1000 steps. For all baseline methods, we employed the optimal set of hyperparameters as indicated in the respective papers. Each task is executed with three different seeds for each model. Detailed explanations of these methods and of the implementations can be found in the Supplementary Material (Section 3).

## 5.3 RESULTS IN STANDARD SETTINGS

The performance of all the evaluated methods in the standard DMC environment is illustrated in the Supplementary Material (Section 5.2). DPI exhibits a degree of effectiveness in certain scenarios involving static backgrounds, although it does not consistently outperform all other methods.

## 5.4 RESULTS IN NATURAL BACKGROUND SETTINGS

Figure 4 illustrates the outcomes when employing natural backgrounds, wherein the background videos were not presented to the agent during its training phase. The main reasons for the degraded performance of most baseline methods was changing the background image to RGB. Dreamer struggles to accurately capture the agent's entire state, and inadvertently incorporates the irrelevant background noise into its representation (Supplementary Material Section 5). TIA, on the other hand, can only effectively distinguish the agent from the distractor when the background is rendered in grayscale. DBC's performance is on par with these methods, however, it does not achieve the performance that was reported by Zhang et al. (2021). This discrepancy is largely due to the inclusion

of RGB image in the background and authors' approach to use the same video for both training and testing, which hampers its capability to manage diverse distraction and restricts its generalization capability to unseen distractions. Similarly, TPC (Nguyen et al., 2021) and Denoised MDPs (Wang et al., 2022) underperformed due to its incapability to generalise to diverse unseen distractions. Our implementation utilises the authors' open-sourced code, with the sole adjustment being the introduction of additional videos. Contrary to these methods, DPI achieves better rewards in the top three environments in Figure 4. This success can be attributed to the state-space model that integrates the actions into the latent representations. Such integration results in a reconstructed scene where the background is blurred, and the agent is enhanced, signifying DPI's capacity to encode task-relevant components, enhancing its performance even in complex and noisy environments (Reconstruction Results in Supp. Material Section 6).

**Failure under Sparse rewards.** As illustrated in Figure 4, our approach excels in Dense reward scenarios (e.g., Cheetah run, Walker walk, Cartpole swingup). However, it struggles with sparse reward environments (Cup Catch and Pendulum Swingup) after $10^6$ environment steps. The complexity of the task, when paired with the visual noise in the environment, presents a considerable challenge and surpasses the limits of current methodologies. In conclusion, the tasks that are inherently hard for model-based methods would remain hard for DPI. Significant improvements can be made for exploration in sparse reward contexts.

### 5.5 RESULTS IN RANDOM BACKGROUND SETTINGS

In this experiment, every time instance features a unique background image, inducing maximum stochasticity in the environment. This experiment illustrates the preservation of temporally predictive information by DPI. As demonstrated in Figure 1, DPI effectively isolates task-relevant features, managing to reconstruct only the agent against a randomized background. Denoised MDPs (Wang et al., 2022), our closest competitor here, exhibits high variance and instability, making its results less reliable and subject to fluctuation, undermining its utility in stochastic environments. Figure 5 further highlights DPI's superior control performance in comparison to all other baselines (More in Supplementary Material Section 6).

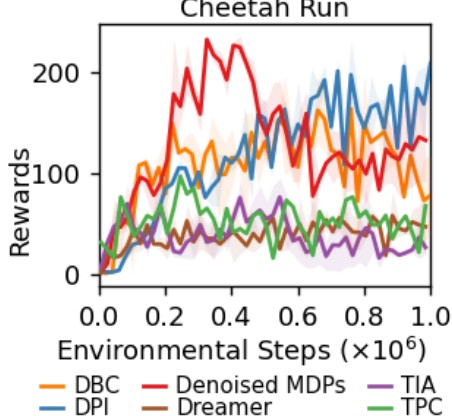

Figure 5: **Random Background Setting**. Comparison of DPI with baselines in random background setting on three runs.

### 6 DISCUSSION AND CONCLUSION

Our work demonstrates that our information-theoretic formulation suggests a pathway to segregate and represent task-relevant information in a noisy world, without explicitly modelling any rules of the MDPs. We also show that objectives related to maximising information on various variables, that are explicitly mentioned in other research (Bai et al., 2021; You et al., 2022; Lee et al., 2020b), implicitly emerge out from our theoretical formulation. In our analysis, all the methodologies exhibit strong performance in noise-free scenarios. When subjected to natural noise scenarios, characterized by real-world videos, DPI consistently either surpassed or equaled the best of eight baselines in performance. However, there's a noticeable path for improvement as every method encountered challenges in tasks dominated by sparse rewards (bottom row of Figure 4). Most notably, in random noise conditions, DPI does not face significant drop in performance and outperforms all other baseline methodologies.

Our method can be combined with any existing RL model that performs exponentially well in noise-free environment. We believe that there is a great room for improving the performance of our model, e.g., by improving the model architecture for the encoding representations using Resnet like in Bai et al. (2021), by utilising experience replay sampling strategies like PER (Schaul et al., 2016), or by incorporating sophisticated exploration strategies for sparse environments (Laskin et al., 2020; You et al., 2022).

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
