# OpenReview forum: "Information-Theoretic World Model learning for Denoised Predictions"
_ICLR.cc/2024/Conference — Submitted to ICLR 2024_

### Official Review · Reviewer_Dg3d · 2023-10-31

**Soundness:** 2 fair
**Presentation:** 2 fair
**Contribution:** 2 fair
**Rating:** 5
**Confidence:** 3

**Summary:**

Rebuttal response: Thank you taking the time to produce a detailed rebuttal, I increased my score.  I still have some of my basic earlier concerns, one being that the experimental results don't seem sufficiently strong.  Perhaps an analytical domain where there's a clearer benefit could help to supplement the more complicated experimental domains.  I'm also concerned about the necessity of adding the extra weighting hyperparameter, especially if results seem to be fairly sensitive to it.  Another big issue is making the motivation for the overall technique clearer.

---

This paper proposes to maximize mutual information between the current state and future state while minimizing mutual information between the past state and current state.  This is then added as an auxiliary loss into the soft actor critic setup, and is used in continuous control tasks with known rewards.  One of my biggest issues with this paper is the empirical weakness of the results, which are inconclusive in the noisy setting, and much worse in the noise-free setting (in the appendix).  The other issue concerns the quality and novelty of the idea.  It feels very similar to the InfoPower paper and it's not obvious that the proposed objective is a good way of removing distractors, nor is it clearly justified.

Notes:
  -Filtering background noise from videos.
  -Minimize past information, keep info about future.
  -SAC with information-based loss.
  -Uses "CLUB" mutual information method to increase MI between current state and future state.

**Strengths:**

-The idea of maximizing mutual information between current/future state while minimizing mutual information between past/current state is intriguing but a bit counterintuitive.
  -Doing noise removal in the online setting seems like an important problem.

**Weaknesses:**

-The reconstructions aren't terribly compelling.  While background info is removed, it's not obvious if all the information about the agent is correctly retained.
  -Papers like ACRO (Islam 2022) and InfoGating (Tomar 2023, which I suppose is a bit recent) should be cited.  InfoPower also feels relevant, and while it is cited, the difference doesn't seem to be sufficiently discussed.
  -The integration with SAC feels overly specific.
  -The experimental results seem fairly weak.  There is a clear improvement in cheetah run and a small improvement in walker walk, but other tasks are more modest.
  -The results in the appendix on standard setup seem quite bad as the proposed method is badly losing to Dreamer.
  -The paper doesn't do much to analyze what is happening in the information bottleneck.

**Questions:**

-The paper refers to the setup as a lagrangian formulation, but for this to work, it seems insufficient to treat beta as a hyperparameter.  So what exactly makes it a lagrangian setup, as opposed to just adding an extra penalty?

  -What is the reason to think that increasing mutual information between present and future states is compatible with removing distractors?  For example, the background video will have temporal correlation, so we could potentially maximize that by keeping the TV in the state.  See Efroni 2022 (PPE) as well as Lamb 2022 (AC-State) for some discussion of this issue.

---

> ### Author Response · Authors · 2023-11-20
> **Response from the authors**
>
> We appreciate your time and feedback. We would also appreciate your concerns on the empirical weakness. We have worked on the issues as follow:
>
> **[W1] The reconstructions aren't terribly compelling. While background info is removed, it's not obvious if all the information about the agent is correctly retained.**\
> Acknowledging the observations about the reconstructions, it is important to highlight from our results that the method retains the majority of the information about the agent. The process effectively darkens the background, predominantly isolating and preserving the agent's details. This indicates that while background information is removed, the critical aspects of the agent are largely maintained. Future iterations of our model will focus on refining this aspect to ensure even more precise retention of all critical information about the agent. For more analysis, we have also added the reconstruction of the Cartpole Swingup task in Figure 5 of the Supplementary Material.
>
> **[W2] Papers like ACRO (Islam 2022) and InfoGating (Tomar 2023, which I suppose is a bit recent) should be cited.**\
> **[A2]** Thanks for bringing these papers to our attention. We have cited both the papers.
>
> **[W3]  InfoPower also feels relevant, and while it is cited, the difference doesn't seem to be sufficiently discussed.**\
> **[A3]** Thank you for pointing out the relevance of InfoPower in our discussion. While InfoPower focuses on maximizing empowerment, specifically $I(a_k;z_{k+1}|z_k)$, our objective diverges significantly from this. We deconstruct $I(z_{k+1},a_k;z_k)$ into two components: $I(z_k;z_{k+1})$ and $I(z_k,a_k|z_{k+1})$. Although the second term might appear similar to InfoPower's approach, it fundamentally differs in both concept and implementation. It's true that InfoPower and other methods we cited [1-3] employ an inverse dynamic model $q(a_k|z_k,z_{k+1})$, but our approach advocates a more generalized strategy. We propose that focusing on $I(z_{k+1},a_k;z_k)$ implicitly encompasses learning inverse dynamics, diverging from InfoPower's explicit modeling of this aspect. We have cited InfoPOWER in Section 4.3 now.
>
>
> **[W4] The integration with SAC feels overly specific.**\
> **[A4]** The detailed description of our SAC integration is crucial for both reproducibility and clear understanding, enabling other researchers to accurately replicate and build upon our work.
>
> **[W5] The experimental results seem fairly weak. There is a clear improvement in cheetah run and a small improvement in walker walk, but other tasks are more modest.**\
> **[A5]** In our rebuttal, we present enhanced results in additional environments such as Walker Walk and Cartpole Swingup, achieved through weighing coefficient tuning conducted subsequent to the paper's initial submission. Our updated results now show a significant improvement over the baseline methods in the specified environments. We encourage reviewing the new graphs presented in Figure 4 for a detailed comparison of these enhancements. The details of these newly optimized weighing coefficient are available in Section 4 of the Supplementary Material. We have also added Cartpole swingup in the random settings (Figure 2 for results and Figure 5 for reconstruction in the Supplementary Material), which clearly aligns with our claims. Additionally, we deliberately highlight the limitations of all methods, including our own, in three challenging environments. This was done to underscore the current limitations of state-of-the-art methods in handling these complex environments, thereby emphasizing the need for further research in this area.
>
> **[W6] The results in the appendix on standard setup seem quite bad as the proposed method is badly losing to Dreamer.**\
> **[A6]** DPI, although not outperforming Dreamer in static environments, is designed for effective generalization across natural settings. As illustrated in the graph, other methods in the same category as DPI also face challenges in these environments. Nevertheless, DPI maintains competitive performance compared to other methods (excluding Dreamer) in four out of the six tested environments.

---

> ### Author Response · Authors · 2023-11-20
> **Response from the authors-II**
>
> **[Q1] The paper refers to the setup as a lagrangian formulation, but for this to work, it seems insufficient to treat beta as a hyperparameter. So what exactly makes it a lagrangian setup, as opposed to just adding an extra penalty?**\
> Thanks for this question. We use the similar terminology utilises in Information Bottleneck Principle (IB) [4] to be uniform with the theory. In IB formulation, this parameter is responsible for the tradeoff between the complexity of representation and  amount of preserved relevant information. Similarly, in our model, these parameters are instrumental in balancing the relationship between past and future variables, with the present variable acting as the bottleneck.
>
> **[Q2] What is the reason to think that increasing mutual information between present and future states is compatible with removing distractors? For example, the background video will have temporal correlation, so we could potentially maximize that by keeping the TV in the state. See Efroni 2022 (PPE) as well as Lamb 2022 (AC-State) for some discussion of this issue.**\
> **[A2]** This is a great question and thanks for suggesting AC-State [5] especially. Our idea can also be viewed in this way: Forward models while adept at predicting future states, can falter in noisy environments as it can also model task-irrelevant details. Inverse dynamic models, on the other hand, are effective in determining actions based on state transitions but can suffer from a similar problem. Our approach, encompasses both the models, learning predictive models ($q(z_{k+1}|z_k,a_k,h_h)$ in Section 4.2) for the future states and learning actions from the inverse dynamic models implicitly with lower bound on $I(z_{k+1},a_k;z_k)$, which acts as a regularizer.
>
> **References**\
> [1] Amy Zhang, Rowan Thomas McAllister, Roberto Calandra, Yarin Gal, and Sergey Levine. Learning invariant representations for reinforcement learning without reconstruction. In International Conference on Learning Representations, 2021. URL https://openreview.net/forum?id=-2FCwDKRREu. \
> [2] Deepak Pathak, Pulkit Agrawal, Alexei A. Efros, and Trevor Darrell. Curiosity-driven exploration by self-supervised prediction. In Doina Precup and Yee Whye Teh (eds.), Proceedings of the 34th International Conference on Machine Learning, volume 70 of Proceedings of Machine Learning Research, pp. 2778–2787. PMLR, 06–11 Aug 2017. URL https://proceedings.mlr.press/v70/pathak17a.html.  \
> [3] Yash Chandak, Georgios Theocharous, James Kostas, Scott Jordan, and Philip Thomas. Learning action representations for reinforcement learning. In International Conference on Machine Learning, pp. 941–950. PMLR, 2019. \
> [4] Tishby, Naftali, and Noga Zaslavsky. "Deep learning and the information bottleneck principle." 2015 ieee information theory workshop (itw). IEEE, 2015.

---

> ### Author Response · Authors · 2023-11-22
>
> Thank you again for your valuable feedback . We have addressed all the questions raised in our rebuttal and are ready to respond to any further queries you have. We eagerly await your updated feedback.

---

### Official Review · Reviewer_cwoe · 2023-10-31

**Soundness:** 2 fair
**Presentation:** 1 poor
**Contribution:** 2 fair
**Rating:** 3
**Confidence:** 4

**Summary:**

The paper purports to introduce a mutual-information based object for learning representations in RL that will be robust to noise. The loss is composed of 4 components that are designed to either maximize or minimize the mutual information of various quantities, with bound provided on each part.  The method performs favorably on some benchmark tasks that involve background noise.

--update--

Thank you for the clarifications on the loss as well as the additional baselines and improved results. Some of the explanations are good (why use two bottlenecks with this model), and others not so great (I'm still concerned about the action conditioning in the two terms which appear to work against each other).

However, my concerns about the story and the motivations of the complex loss still remain, and I would have hoped for a revision that at least addressed these issues. I acknowledge that you disagree on how the story is laid out with respect to the problem, which is a bit of a mess, and the relation to prior works is not well laid out. On the second point, my main complaint is how these are presented: it is wholly unclear what DPI inherits (and the wording seems to hedge on this inheritance, opting for "related to") and which models represent an ablation of these different components. In the current reading (which is very close to the original), the paper still suffers quite a bit in this regard. So I'm still at this point hesitant to increase my score, since the paper appears to want it both ways: novelty in terms of the individual components and their combination and relationship in terms of the works that do similar things. If all of these losses are related to these prior works: why not just combine them into one loss function?

**Strengths:**

Using mutual information as a "grounding principle" for representation learning has some strong precedent given prior works, the ease of theoretical interpretation, and existing tractable methodology. The problem is well grounded as "avoiding nuisance variables" in representation learning is important (e.g., for identifiability / generalization). The results are not bad and demonstrate some of the ideas here work.

**Weaknesses:**

Unfortunately, the paper has several issues that cannot be overlooked.

First and foremost, much of the story is a bit sloppy and even problematic at places: the introduction seems to claim without much support that RL only works well when the observations do not contain noise variables that aren't related to the task at hand. There's sort of a skimming over the real problem, particularly in POMDP settings, where reward-relevant signal might be particularly sparse in observations, which can hurt the sample complexity of algorithms in many problems. The problem is often multifaceted, as the SNR (wrt task relevant factors, eg are the task variables themselves noised), reward sparsity, the degree of which the environment is partially observed, etc all factor in towards the success of various algorithms. But overall I'm not sure why, given the story of the introduction, that this has anything to do with task relevance in the representations: learning all of the transition function, even the noisy part, still should be decodable to a policy that can maximize returns. The problem is that these representations might not generalize well (e.g., when the noise correlates with a subset of task-relevant variables).

Second, the components of the loss are a bit complex and not well motivated. Some of the components seem to be doing similar things, e.g., it's unclear whether the two bottleneck terms need to both be included since all the messaging needs to be passed through z_t anyways. Why are all of these terms needed and what do they do that is distinct from the others?

Some of the bounds are fine and rather standard, but it's unclear whether L_CLUB is doing a good thing or not: since we've now introduced the conditional on the actions on the top, it seems like this will discourage the model from remembering which actions were taken in the past. Wouldn't this do the opposite as stated is a goal for this model: to encode controllable, task-relevant information? But then we have I_LTC which has actions as the conditioning variable on the top of the upper bound: it seems like this and the action conditioning part of I_CLUB are doing opposite things?

There is a missing baseline which I think is very important and should be tested agains: Self-Predictive Representations (SPR).

But related to baselines (including SPR), what is really missing is what precisely all of the loss components of this model has are either 1) present in existing baselines or 2) not present in existing baselines. The overall claim is this paper is contributing through MI, but this has been done many times before, just not in the precise form. So much more work needs to be done to ground the contributions wrt the loss components in prior works, as well as better motivating these as mentioned above. And more to the point it's unclear *why* having a MI based objective of this form(s) is necessary: there are other ways to implement MI maximizations / minimizations approximately (eg regression losses as in SPR) that may just work better.

Finally, the results are a bit lackluster. On some experiments (e.g., cheetah run) things go very well, but others it's not so clear (most in fact). This makes me concerned that some bit of hyper parameter tuning was performed on half cheetah that wasn't done on other environments (or baselines).

**Questions:**

Most of my questions are as concerns / weaknesses above.

---

> ### Author Response · Authors · 2023-11-20
> **Response from the authors**
>
> First and foremost, we would like to express our sincere appreciation for the insightful review and valuable feedback on our paper. Your detailed questions have prompted crucial clarifications, enhancing the overall depth and quality of our work. Additionally, your mention of a relevant method has significantly contributed to strengthening the empirical aspects of our paper.
>
> **[W1] First and foremost, much of the story is a bit sloppy ... task-relevant variables).**\
> **[A1]** It is crucial to clarify the narrative presented in the introduction. The assertion that reinforcement learning (RL) faces challenges in noisy environments is not intended to imply that RL is only effective in the absence of noise. Rather, the point is to highlight the increased complexity and potential for sub-optimal performance when RL algorithms encounter observations cluttered with irrelevant information. Regarding POMDPs and reward sparsity, the reviewer's point is well-taken. Our focus on noise and irrelevant information in the introduction is meant to complement, not overlook, these challenges. The introduction's mention of task relevance in representations is to underline the importance of extracting meaningful information from noisy data, not to suggest that the other variables are insignificant fot the success of a task. By emphasizing task relevance in representations, we underline the importance of extracting meaningful information from noisy data, crucial for ensuring generalizable and robust model performance. The introduction thus sets the stage for presenting Denoised Predictive Imagination (DPI) as a method, designed to tackle these challenges in RL through an information-theoretic approach.
>
> **[W2] Second, the components of the loss are a bit complex and not well motivated. Some of the components seem to be doing similar things, e.g., it's unclear whether the two bottleneck terms need to both be included since all the messaging needs to be passed through z_t anyways. Why are all of these terms needed and what do they do that is distinct from the others?**\
> **[A2]** Our core concept revolves around the bottleneck variable $z_t$, which emerges from two distinct processes outlined in Section 4. One is to learn representation from the observations and vice-versa i.e. $o_t \rightarrow z_t$ and $z_t \rightarrow o_t$ respectively. The other process occurs in the latent space itself between the representations i.e. from one representation to another ($z_{t-1} \rightarrow z_t \rightarrow z_{t+1}$). Thus, this creates a two-fold bottleneck. In equation 1, intuitively, the left part $I(o^-,z^-)$ refers that the variable z is being inferred from the observation in a compressed form, while the right part $I(o^+,z^+)$ aims at maximally retaining the information from $z^+$ to predict the subsequent observations $o^+$. This same logic applies in equation 2.
>
> **[W3] Some of the bounds are fine and rather standard, but it's unclear whether L_CLUB is doing a good thing or not: since we've now introduced the conditional on the actions on the top, it seems like this will discourage the model from remembering which actions were taken in the past. Wouldn't this do the opposite as stated is a goal for this model: to encode controllable, task-relevant information? But then we have I_LTC which has actions as the conditioning variable on the top of the upper bound: it seems like this and the action conditioning part of I_CLUB are doing opposite things?**\
> **[A3]** The reviewer's concern about the conditional on actions in $L_{CLUB}$ potentially discouraging the model from remembering past actions is insightful. However, the intention behind $L_{CLUB}$ is to enhance the model's focus on the state transitions, rather than diminishing its recall of past actions. It utilises the variable $h_k$, which is a histocial variable, incorporating information from the past.  $I_{LTC}$, with its action conditioning, assures that inferring $z_{k+1}$ from $o_{k+1}$ should diverge minimally from the $z_{k+1}$ obtained with $z_k$.
>
> **[W4] There is a missing baseline which I think is very important and should be tested agains: Self-Predictive Representations (SPR).**\
> **[A4]** Thank you for pointing out the omission of Self-Predictive Representations (SPR) [1] as a baseline in our study. Following your suggestion, we have now included SPR as a baseline in our experiments. In the Figure 4, we show that DPI significantly outerpforms SPR in Cheetah run, Walker walk and Cartpole swingup. In Figure 2 of the Supplementary Material, we have also shown that DPI outperforms SPR significantly in the cartpole swingup task in random settings. Additionally, we have also added Dreamer-V2 [2] and VSG [3] as a baseline, which significantly contributed to improving the quality of our research.

---

> ### Author Response · Authors · 2023-11-20
> **Response from the authors-II**
>
> **[W5] But related to baselines (including SPR) ... work better.**\
> **[A5]** Thank you for addressing this. We have consistently mentioned many different baselines in each section representing every component. For example, in Section 4.3, while addressing $I(z_{k+1},a_k;z_k)$, we have mentioned that our objective "is closely related to the predictive coding objective [4,5]". On the top of that, while splitting the term, we have mentioned that the inverse dynamic model resembles to [6-8]. Again, in the same section, while working on $I(z^+,o^+)$, we said that we optimise the lower bound on our MI objective similar to [9]. We respectfully disagree from the reviewer regarding not mentioning the prior work in our loss formulations. The primary focus of our paper is to introduce a novel and generalised framework based on Mutual Information, designed to minimize noise and extract task-relevant information from observations. It is indeed true that there are multiple ways to implement MI maximizations or minimizations, such as using regression losses as seen in Self-Predictive Representations (SPR). Our current approach is designed due to its tight bound and ease in implementation, but we are open to exploring other methodologies in future research.
>
> **[W6] Finally, the results are a bit lackluster. On some experiments (e.g., cheetah run) things go very well, but others it's not so clear (most in fact). This makes me concerned that some bit of hyper parameter tuning was performed on half cheetah that wasn't done on other environments (or baselines).**\
> **[A6]** We appreciate your observation regarding the performance of DPI in natural background settings. In our rebuttal, we present enhanced results in additional environments such as Walker Walk and Cartpole Swingup, achieved through setting correct weighing coefficients, conducted subsequent to the paper's initial submission. The details of these newly optimized weights are available in Section 4 of the Supplementary Material. With these optimization, DPI now shows significantly better performance in environments such as Cheetah Run, Walker Walk, and Cartpole Swingup, aligning more closely with our initial claims (Figure 4). We also show a dominant performance of DPI w.r.t. SPR and Dreamer-V2 in cartpole swingup task in random environment (Figure 2 Supplementary Material). On the top of that, we also showed the reconstruction results in the Figure 5 of the Supplementary Material.
>
> **References**\
> [1] Schwarzer, Max et al. “Data-Efficient Reinforcement Learning with Self-Predictive Representations.” International Conference on Learning Representations (2020).\
> [2] Hafner, D.; Lillicrap, T. P.; Norouzi, M.; Ba, J. Mastering Atari with Discrete World Models. International Conference on Learning Representations. 2021\
> [3] Jain, Arnav Kumar, et al. "Learning robust dynamics through variational sparse gating." Advances in Neural Information Processing Systems 35 (2022): 1612-1626. \
> [4] Aaron van den Oord, Yazhe Li, and Oriol Vinyals. Representation learning with contrastive predictive coding. arXiv preprint arXiv:1807.03748, 2018. \
> [5] Ankesh Anand, Evan Racah, Sherjil Ozair, Yoshua Bengio, Marc-Alexandre Côté, and R Devon Hjelm. Unsupervised state representation learning in atari. Advances in Neural Information Processing Systems (NeurIPS), 32, 2019.\
> [6] Amy Zhang, Harsh Satija, and Joelle Pineau. Decoupling dynamics and reward for transfer learning, 2018. URL https://openreview.net/forum?id=H1aoddyvM. \
> [7] Deepak Pathak, Pulkit Agrawal, Alexei A. Efros, and Trevor Darrell. Curiosity-driven exploration by self-supervised prediction. In Doina Precup and Yee Whye Teh (eds.), Proceedings of the 34th International Conference on Machine Learning, volume 70 of Proceedings of Machine Learning Research, pp. 2778–2787. PMLR, 06–11 Aug 2017. URL https://proceedings.mlr.press/v70/pathak17a.html. \
> [8] Homanga Bharadhwaj, Mohammad Babaeizadeh, Dumitru Erhan, and Sergey Levine. Information prioritization through empowerment in visual model-based RL. In International Conference on Learning Representations, 2022. URL https://openreview.net/forum?id=DfUjyyRW90. \
> [9] Alexander A. Alemi, Ian Fischer, Joshua V. Dillon, and Kevin Murphy. Deep variational information bottleneck. In International Conference on Learning Representations, 2017. URL https://openreview.netforum?id=HyxQzBceg.

---

> ### Author Response · Authors · 2023-11-22
>
> Thank you again for your valuable feedback . We have addressed all the questions raised in our rebuttal and are ready to respond to any further queries you have. We eagerly await your updated feedback.

---

### Official Review · Reviewer_qnYC · 2023-11-01

**Soundness:** 3 good
**Presentation:** 3 good
**Contribution:** 3 good
**Rating:** 5
**Confidence:** 4

**Summary:**

The authors propose a novel information-theoretic world model through leveraging the principle of information bottleneck. It can distill task-relevant information for policy learning in noise-saturated environments.

**Strengths:**

1. This paper focused on unexplored noise-saturated scenarios and proposed a new methed, Denoised Predictive Imagination, to tackle such situations.
2. A detailed derivation of Denoised Predictive Imagination was given in the paper.
3. DPI outperformed five state-of-the-art baseline models on six modified DeepMind control tasks.

**Weaknesses:**

1. The design of experiments can be improved. Switching background is not convincing enough to demonstrated the effectiveness of the proposed method.

2. Other basline model like DreamerV2 and DreamerV3 can also be included for comparison.

**Questions:**

I will increase my rating if all of my concerns are properly addressed.

Thank authors' for their detailed rebuttal, hovewer, not all of my concerns are well addressed. I will keep my score unchanged.

---

> ### Author Response · Authors · 2023-11-19
> **Response from the authors**
>
> First and foremost, we express our sincere appreciation for the review and valuable insights provided on our paper. Your feedback has played a significant role in shaping our research, and we are very grateful for it.
>
> **[W1] The design of experiments can be improved. Switching background is not convincing enough to demonstrated the effectiveness of the proposed method.**\
> **[A1]** Thank you for your feedback. We appreciate your perspective on the experimental design. It's important to emphasize that changing backgrounds presents one of the most challenging problems in this field, and is an indicator of generalization and robustness. Our experiments, designed around this challenge, are intended to specifically address and demonstrate the effectiveness of our proposed method in tackling this complex issue. We also highlight how other models, despite their strong performance in static backgrounds, fail to maintain their effectiveness in these more challenging dynamic background settings. This contrast not only underscores the unique robustness of our proposed method but also reinforces the complexity and significance of the problem we are addressing.
>
>
> **[W2] Other basline model like DreamerV2 and DreamerV3 can also be included for comparison.**\
> **[A2]** Thank you for suggesting these methods. We have included DreamerV2 in our comparative analysis, but due to time and resource constraints, we were unable to incorporate DreamerV3. This decision was made to ensure a thorough and focused evaluation within our available means. Additionally, we have incorporated other relevant methods as suggested by other reviewers, SPR [1] and VSG [2], broadening the scope of our analysis. Notably, our method outperforms all these included models, showcasing its efficacy and robustness in the evaluated scenarios.
>
> **Additional Comments:**
> - Following suggestions from other reviewers, we have incorporated these additional baseline methods, SPR [1] and VSG [2], into our analysis.
> - Our updated results now show a significant improvement over the baseline methods in the specified environments. We encourage reviewing the new graphs presented in Figure 4 for a detailed comparison of these enhancements.
> - Additional experiments have been conducted on the Cartpole Swingup task within random environments. These new results clearly demonstrate DPI's capability to filter out background noise and effectively extract relevant information (Results in Supplementary Material Figure 2 and Reconstruction in Supplementary Material Figure 5).
>
> **References**\
> [1] Schwarzer, Max et al. “Data-Efficient Reinforcement Learning with Self-Predictive Representations.” International Conference on Learning Representations (2020).\
> [2] Jain, Arnav Kumar, et al. "Learning robust dynamics through variational sparse gating." Advances in Neural Information Processing Systems 35 (2022): 1612-1626.

---

> ### Author Response · Authors · 2023-11-21
>
> Thank you again for your valuable feedback and for considering an increase in your rating upon the completion of the suggested changes. As the deadline for rebuttal is approaching, we would greatly appreciate receiving any further comments or responses from you. We would like to inform you that we have implemented all the necessary changes suggested by you, including some additional modifications that we believe further strengthen the paper. We appreciate your dedication to the review process and look forward to your prompt feedback on these latest revisions.

---

### Official Review · Reviewer_7DVA · 2023-11-03

**Soundness:** 3 good
**Presentation:** 3 good
**Contribution:** 2 fair
**Rating:** 6
**Confidence:** 3

**Summary:**

In this paper the authors present a novel approach to reinforcement learning problems in noisy environments through a world model that learns denoised predictions of the environment. The effect is essentially like an encoder or a dimensionality reducer, using a set of models to identify only relevant features from the environment. After compression, these features are then used to reconstruct the state in terms of only relevant features, which is then used to guide an agent. The paper presents results across three cases, finding that their approach meets or exceeds the performance of relevant baselines.

**Strengths:**

In terms of clarity, the paper does a good job making an argument for the importance of this problem and the authors' approach to solving it. The technical details are somewhat dense, but exhaustive which helps clearly express the approach. The setup for the experiments is also very clear.

In terms of quality, the technical quality of the work is high. The detail in the paper is sufficient for reproducibility concerns, which is further strengthened with the supplementary material. There don't appear to be major issues with any equations or figures.

In terms of originality, while there are prior work in terms of denoising and reinforcement learning, the authors do a good job of overviewing it and differentiating their approach.

In terms of significance, the paper would certainly be of interest to researchers interested in RL for noisy environments.

**Weaknesses:**

I identify one major and one minor weakness in the current paper draft.

The major weakness is in the experiment setups, results, and discussion around them. In the paper, the authors refer to the results of the first experiment with the static backgrounds as "competitive with all other methods". This is perhaps a bit generous, when DPI as below median performance in four of the six settings. For the natural background settings, the claim is that "DPI achieves better rewards in nearly all environments", which may be true in terms of the blue line spinning above the others, but the performance of DPI only outperforms the baselines in two of the six tasks (Cheetah Run and Walker Walk). Finally, for the random background setting, results are only given for DPI's best performing environment (Cheetah Run). Cheetah Run is also the only environment in the visualized examples, which further supports that it is DPI's best performing environment. Beyond the text perhaps overstating the performance of DPI, there's also no discussion of why the performance of DPI should improve so much in this one environment compared to the others or why it's appropriate to only showcase visualizations and random background performance in it. These issues currently lead to me, as a reader, wondering if the complexity of DPI is justified by the relatively small number of environments it improves in, compared to baselines. Clarification on this point would be greatly appreciated.

In terms of the minor weakness, the technical approach section is very dense, making it difficult to get a clear sense of the high level structure of the approach. Figure 2 exists, but is a model of the problem without a strong connection to the implemented solution. Figure 3 also exists but only covers two of the models in the approach. A complete pipeline diagram or text overview would be greatly appreciated.

**Questions:**

1. Why does DPI perform so much better in Cheetah Run?
2. Why does DPI's performance in the static environments perform below the median in so many cases?
3. Why only include visualizations and random background results in Cheetah Run? What did the other environments look like?

---

> ### Author Response · Authors · 2023-11-19
> **Response from the authors**
>
> First and foremost, we express our sincere appreciation for the review and valuable insights provided on our paper. Your feedback has played a significant role in shaping our research, and we are very grateful for it.
>
> **[W1]  In the paper, the authors refer to the results of the first experiment with the static backgrounds as "competitive with all other methods". This is perhaps a bit generous, when DPI as below median performance in four of the six settings.**\
> **[A1]** Thank you for highlighting this aspect. We have revised the statement to reflect a more nuanced understanding: "DPI exhibits a degree of effectiveness in certain scenarios involving static backgrounds, although it does not consistently outperform all other methods."
>
> **[W2] For the natural background settings, the claim is that "DPI achieves better rewards in nearly all environments", which may be true in terms of the blue line spinning above the others, but the performance of DPI only outperforms the baselines in two of the six tasks (Cheetah Run and Walker Walk).**\
> **[A2]** We appreciate your observation regarding the performance of DPI in natural background settings. In our rebuttal, we present enhanced results in additional environments such as Walker Walk and Cartpole Swingup, achieved through hyperparameter tuning conducted subsequent to the paper's initial submission. The details of these newly optimized hyperparameters are available in Section 4 of the Supplementary Material. With these optimizations, DPI now shows significantly better performance in environments such as Cheetah Run, Walker Walk, and Cartpole Swingup, aligning more closely with our initial claims (Figure 4).
>
> **[W3] These issues currently lead to me, as a reader, wondering if the complexity of DPI is justified by the relatively small number of environments it improves in, compared to baselines. Clarification on this point would be greatly appreciated.**\
> **[A3]** As mentioned above, DPI now shows significantly better performance in environments such as Cheetah Run, Walker Walk, and Cartpole Swingup, aligning more closely with our initial claims (Figure 4).
>
> **[W4] Finally, for the random background setting, results are only given for DPI's best performing environment (Cheetah Run).**\
> **[A4]** Thank you for pointing out the scope of our results in the random background setting, particularly focusing on DPI's performance in Cheetah Run. Our approach for experimental evaluation in random settings was modeled after the TPC's [1] method. The primary constraint for the limited range of experiments was computational cost and tight timeline. It was challenging to conduct comprehensive experiments across all tasks, including comparisons with baselines. However, we managed to extend our analysis to include the Cartpole Swingup environment within the random background settings, offering a broader view of DPI's performance. The results in Figure 2 in Supplementary Material shows that DPI outperforms the two selected baselines significantly.
>
> **[W5] These issues currently lead to me, as a reader, wondering if the complexity of DPI is justified by the relatively small number of environments it improves in, compared to baselines. Clarification on this point would be greatly appreciated.**\
> **[A5]** The primary focus of our paper is to introduce a novel framework based on Mutual Information, designed to minimize noise and extract task-relevant information from observations. This approach, particularly the use of predictive information for noise reduction, is a novel contribution to the field. As we have highlighted, following further optimization, DPI now demonstrates significantly improved performance in environments such as Cheetah Run, Walker Walk, and Cartpole Swingup. This enhancement in performance aligns more closely with our initial claims and underscores the efficacy of our approach in these specific settings, thereby justifying the complexity of DPI in the context of our study's objectives.

---

> > ### Comment · Reviewer_7DVA · 2023-11-19
> > **Re: Response from the authors**
> >
> > Thank you for taking the time to address my concerns and questions! However, I am somewhat worried that the solution appears to have been to find new hyperparameters that supported the original claims. The hyperparameter tuning process also does not appear to be covered anywhere. This raises concerns similar to training on the test set, and related concerns around the generalizability of the framework. Can the authors clarify the nature of the hyperparameter tuning process?

---

> > > ### Author Response · Authors · 2023-11-19
> > > **Response to the reviewer**
> > >
> > > Thank you for your prompt response and highlighting this issue. We regret the error in our explanation and apologize for any confusion caused. What we intended to describe was the "weighing coefficients" of various loss functions. This has now been corrected and updated in Section 4.3 of the Supplementary Material (Table 2). All the other factors remained constant.
> > >
> > > Regarding the concern about training and testing on the same set, we would like to clarify that we employed a random split of the Kinetic-400 Dataset into 100 videos each for the training and test sets. This approach ensures that the videos in the training set are distinct from those in the test set. This was performed similarly on all the mentioned baselines. We appreciate your attention to this detail and hope this clarification addresses your concern.

---

> > > > ### Comment · Reviewer_7DVA · 2023-11-21
> > > > **Re: Response to the reviewer**
> > > >
> > > > Thanks for the quick response and clarification! Yes, that fully addresses that concern. I will increase my rating to reflect this. However, I now share some of the concerns of my fellow reviewers and I would want to see whether they felt their concerns had been addressed before any further increases.

---

> > > > > ### Author Response · Authors · 2023-11-21
> > > > >
> > > > > Thank you very much for your prompt response and for adjusting your rating based on the clarifications provided. We are also very keen to address any additional concerns or questions you might have, particularly in relation to points raised by other reviewers.

---

> ### Author Response · Authors · 2023-11-19
> **Response from the authors-II**
>
> **[Q1] Why does DPI perform so much better in Cheetah Run?**\
> **[A1]** We have conducted hyperparameter tuning across all environments. Please refer to the updated results for further information (Figure 4, Figure 2 in Supplementary Material).
>
> **[Q2] Why does DPI's performance in the static environments perform below the median in so many cases?**\
> **[A2]** DPI, although not outperforming Dreamer in static environments, is designed for effective generalization across natural settings. As illustrated in the graph, other methods in the same category as DPI also face challenges in these environments. Nevertheless, DPI maintains competitive performance compared to other methods (excluding Dreamer) in four out of the six tested environments.
>
> **[Q3] Why only include visualizations and random background results in Cheetah Run? What did the other environments look like?**\
> **[A2]** We have added Cartpole environment in the Supplementary Material (Figure 5). However, due to time constraints for the rebuttal, we were unable to compare it with all the baselines. Thus, we have compared it with the two methods (Supplementary Material Figure 2) suggested by other reviewers: SPR [2] and Dreamer-V2 [3].
>
>
> [1] Nguyen, Tung D., et al. "Temporal predictive coding for model-based planning in latent space." International Conference on Machine Learning. PMLR, 2021.\
> [2] Schwarzer, Max et al. “Data-Efficient Reinforcement Learning with Self-Predictive Representations.” International Conference on Learning Representations (2020).\
> [3] Hafner, D.; Lillicrap, T. P.; Norouzi, M.; Ba, J. Mastering Atari with Discrete World Models. International Conference on Learning Representations. 2021

---

### Author Response · Authors · 2023-11-20
**General Comment**

Thank you to all the reviewers for dedicating your time and effort in reviewing our work. Your insightful comments and constructive critiques have not only been invaluable in enhancing the clarity and depth of our research but have also significantly contributed to improving the overall quality of our paper. Here, we summarize the important points and the necessary steps taken by us to resolve them:

**Main Concerns**
1. Empirical Results (**All**)
2. More Baselines (**qnYC**, **cwoe**, **Dg3d**)
3. Moderating the Paper's Tone (**7VDA**)
4. Citation of existing baselines (**cwoe**, **Dg3d**)

**Addressing the concerns**
1. Adding three elements:
   - New results for natural settings (Figure 4) from adjusting weighing coefficient.
   - Adding more environment (Catrpole swingup) for random setting (Figure 2 in Supplementary Material).
   - Adding its reconstructed images (Figure 5 in Supplementary Material) for ensuring noise removal and denoised prediction (**Dg3d**).
2. More Baselines:
   - Total 8 baselines now.
   - Added three more baselines: Dreamer-V2 [1] (**qnYC**), SPR [2] (**cwoe**) and VSG [3].
   - DPI outperforms all the baselines (including the suggested) in the experiments.
3. Moderating the Paper's Tone:
   - Rephrased the mentioned sentences in **7VDA's** rebuttal.
4. Citation of existing baselines:
   - Mentioned paper (InfoPOWER) in correct place.
   - Answered queries from the reviewers (**cwoe**, **Dg3d**) as we already mentioned which loss function resemble to which prior existing method.

**References**\
[1] Hafner, D.; Lillicrap, T. P.; Norouzi, M.; Ba, J. Mastering Atari with Discrete World Models. International Conference on Learning Representations. 2021. \
[2] Schwarzer, Max et al. “Data-Efficient Reinforcement Learning with Self-Predictive Representations.” International Conference on Learning Representations (2020). \
[3] Jain, Arnav Kumar, et al. "Learning robust dynamics through variational sparse gating." Advances in Neural Information Processing Systems 35 (2022): 1612-1626.

---

### Author Response · Authors · 2023-11-23
**Feedback request**

Thank you for your time and effort in evaluating our submission. As we near the conclusion of the discussion phase, we are eager to receive any feedback you might have regarding our rebuttal.

---

### Comment · Area_Chair_sC26 · 2023-11-23
**From AC at the end of rebuttal: Reviewer response required**

Dear Reviewers,

Thanks for your time and commitment to the ICLR 2024 review process.

As we approach the conclusion of the author-reviewer discussion period (Wednesday, Nov 22nd, AOE), I kindly urge those who haven't engaged with the authors' dedicated rebuttal to please take a moment to review their response and share your feedback, regardless of whether it alters your opinion of the paper.

Your feedback is essential to a thorough assessment of the submission.

Best regards,

AC

---

### Meta-Review · Area_Chair_sC26 · 2023-12-10

**Metareview:**

This paper presents an information-theoretic framework for world model learning, aiming at simultaneously learning control policies and at producing denoised predictions. The paper received generally negative initial reviews, but the authors had performed an effective and informative rebuttal, based on which most reviewers increased their scores. However, two reviewers still had crucial concerns. One is that the experimental results don't seem sufficiently strong, with additional concern on adding extra weighting hyperparameters. As a common practice experience, the world model is very sensitive to these hyperparameters, and simply tweaking these hyperparameters may yield very different results, which may impose an unfair evaluation protocol for the baselines (e.g. DreamerV2). The other is that the motivation for the overall technique shall be made clearer. Why the complex loss is a must and how each loss differs from the previous works? The paper needs to undergo a major revision towards addressing these concerns.

**Justification For Why Not Higher Score:**

The empirical performance is not strong while the methodological contribution lacks strong motivation and some of the loss terms lack clear distinction from previous works.

**Justification For Why Not Lower Score:**

N/A

---

### Decision · Program_Chairs · 2024-01-16

Reject